# Remarkable Single Atom Catalyst of Transition Metal (Fe, Co & Ni) Doped on C_2_N Surface for Hydrogen Dissociation Reaction

**DOI:** 10.3390/nano13010029

**Published:** 2022-12-21

**Authors:** Ahmed Bilal Shah, Sehrish Sarfaraz, Muhammad Yar, Nadeem S. Sheikh, Hassan H. Hammud, Khurshid Ayub

**Affiliations:** 1Department of Chemistry, COMSATS University Islamabad, Abbottabad Campus, Abbottabad 22060, Pakistan; 2Chemical Sciences, Faculty of Science, Universiti Brunei Darussalam, Jalan Tungku Link, Gadong BE1410, Brunei; 3Department of Chemistry, College of Science, King Faisal University, Al-Ahsa 31982, Saudi Arabia

**Keywords:** hydrogen dissociation reaction, hydrogen energy, C_2_N surface, single atom catalyst, catalysis, transition metals

## Abstract

Currently, hydrogen is recognized as the best alternative for fossil fuels because of its sustainable nature and environmentally friendly processing. In this study, hydrogen dissociation reaction is studied theoretically on the transition metal doped carbon nitride (C_2_N) surface through single atom catalysis. Each TMs@C_2_N complex is evaluated to obtain the most stable spin state for catalytic reaction. In addition, electronic properties (natural bond orbital NBO & frontier molecular orbital FMO) of the most stable spin state complex are further explored. During dissociation, hydrogen is primarily adsorbed on metal doped C_2_N surface and then dissociated heterolytically between metal and nitrogen atom of C_2_N surface. Results revealed that theFe@C_2_N surface is the most suitable catalyst for H_2_ dissociation reaction with activation barrier of 0.36 eV compared with Ni@C_2_N (0.40 eV) and Co@C_2_N (0.45 eV) complexes. The activation barrier for H_2_ dissociation reaction is quite low in case of Fe@C_2_N surface, which is comparatively better than already reported noble metal catalysts.

## 1. Introduction

Catalysts are the backbone of many commercially available energy conversion and industrial processes [1]. Currently, catalytic technology is managing the production of approximately more than ten trillion dollars of goods annually in power, petroleum, food, and chemicals industries [2]. Noble metals have displayed exceptionally outstanding catalytic properties for energy conversion and production and have surpassed all other catalysts. Despite their wide-range use, there are some challenges associated with noble metals catalysts, such as their limited availability and high prices [3]. From economical perspective, the cost and scarcity of many promising catalytic metals such as palladium and platinum reduces their extensive use [4,5]. To deal with this problem effectively, many other catalysts are being considered with the motive of obtaining commercially viable economical catalyst with uncompromised catalytic efficiency. 

The key goal of alternative approaches being considered is to reduce the quantity of these expensive noble metals while possibly improving or at least maintaining performance. Thus, in recent times, single atom catalysis (SAC) approach has been proposed, so that catalysis can be accomplished via single metal atom on a support surface [6,7,8]. Single atom catalysts usually include uniformly distributed catalyst on supports which act to stabilize these catalysts. With the passage of time, many synthetic procedures have been developed to prepare these catalysts. Novel techniques that are used to synthesize and characterize single atom catalyst are wet chemistry [9], atomic layer deposition [10] and mass selected soft landing [11]. The catalytic efficiency of these catalysts is taking hold, but the understanding of these type of catalysts is still limited. Therefore, the study of supported single metal atom catalysis is of great interest [12].

Over the years, hydrogen has been recognized as the best alternative for fuels because of its sustainable nature and environment friendly processing [13]. Among chemical reactions being catalyzed in industries, hydrogen dissociation reaction is the most carried out process. It is a part of many important chemical reactions including production of ethylene from hydrogen-based energy fuel cells [14], and in Fischer-Tropsch process [15]. It is also part of famous Haber Bosch process for ammonia synthesis [16].

Previous studies have revealed that numerous noble metals such as Pt [17], Ru [18], Pd [19], Rh [20], and Au [21] have the potential to efficiently catalyze hydrogen dissociation reactions. However, these noble metals are very expensive and generally work at high temperature, which make them economically non-feasible [22]. Transition metals such as Mn, Fe, Ni, Co, Zn, and Cu etc. have gained much interest due to their relatively high abundance and low cost [23,24]. Replacing noble metals with a low-cost material is necessary for large-scale and practical application. On the contrary side, the carbon-based materials, organic frameworks [25], graphyne [26], graphene [27], graphitic carbon nitride (g-C_3_N_4_) [28], porous and other nanostructured materials have received considerable attention as adsorbent (support materials for catalyst) due to their promising H_2_ storage capacities, large surface areas at low temperature and potential thermal and electronic properties [29,30]. Yan et al. investigated the Pd doped graphene surface for hydrogen dissociation reaction during the hydrogenation of 1,3-butadiene. They observed that dissociation of H_2_ occurred over the Pd atom (for subsequent hydrogenation) with moderate energy barrier of 0.84 eV [31]. However, scientists are still trying to search and design more efficient catalysts that could offer more selectivity and economical way for hydrogen dissociation reaction.

Recently, a novel 2-D C_2_N monolayer was synthesized experimentally by Mahmood et al. [32]. The special nitrogenated cavities in the C_2_N monolayer are evenly distributed which serve as optimal site for capturing and holding single metal atom. The surface used is quite stable and employed in many research areas including sensors [33], storage materials [34], as support [35] and in drug delivery [36]. Furthermore, metal coordinated C_2_N-based materials are also employed as catalyst in lithium sulfur batteries as well as in oxygen reduction [37,38]. Therefore, the C_2_N sheet can be employed as a support for single metal atom similar to g-C_3_N_4_ and graphene [35]. 2D C_2_N surface has been previously reported in literature to effectively catalyze several important reactions, for example H_2_ evolution reaction, N_2_ reduction, CO [39] and O_2_ reduction reactions [40] due to rich nitrogen content and periodic porous structure [41,42,43,44]. 

Herein, we have investigated the hydrogen dissociation reaction through late transition metal (TM) atoms based single atom catalysis using density functional theory. In the current study, we mainly focus on the dissociation of H_2_ on low-cost transition metals (TMs), for example Fe, Co and Ni supported on C_2_N surface [45,46,47]. Being quite stable, C_2_N surface stabilizes the single atom catalyst quite amazingly. In comparison to carbon-based surface, this surface also bears nitrogen which can play a good role in hydrogen dissociation reaction due to electronegativity difference with hydrogen and can tune the electronic properties of TMs [48,49].

## 2. Computational Methodology

In this study, M06-2X/6-31G(d,p) is used for the optimization of all the structures using Gaussian09 software. M06-2X is a long-range functional which is well reported to estimate non-covalent interaction energies and barrier heights [50]. Benchmark studies show that M06-2X functional perform better for interactions of stacked system. M06-2X also shows better performance when interactions of TMs are studied with other systems [51].

TMs show various spin states, therefore, each of these metal complexes were optimized at various spin states to obtain the most stable spins state. The electronic configurations of Fe, Co and Ni are [Ar] 3d6^4^s^2^, [Ar] 3d6^7^s^2^ and [Ar] 3d6^8^s^2^, respectively. In case of Fe and Ni doped C_2_N, singlet, triplet, quintet, and septet spin states were considered, and the most stable spin states are septet and triplet for Fe and Ni, respectively (see Appendix A). For Co doped C_2_N, doublet, quartet, sextet, and octet spin states were optimized in search of the most thermodynamically stable spin state of metal doped C_2_N. Among the optimized spin states, doublet is the most stable in case of Co@C_2_N complex. The interactions energies of these complexes were based on the stable spin states. Hydrogen dissociation reaction was also performed on the most stable spin state metals. 

Interaction energies are calculated for the most stable geometries of studied complexes by using the following expression: E_int_ = E_T.M@C2N_ − (E_C2N_ + E_T.M_)(1)

Here, E_T.M@C2N_, E_C2N_ and E_T.M_ are interaction energies of TMs@C_2_N complexes, bare C_2_N surface and TMs, respectively.

For hydrogen dissociation, primary step is adsorption of hydrogen on metal doped C_2_N surface. The adsorption energy of hydrogen doped catalyst is calculated through following equation:E_ads._ = E_H2@C2N-M_ − (E_M@C2N_ + E_H2_)(2)

Here, E_H2@C2N-M_, E_M@C2N_ and E_H2_ represent the energies of the hybrid structures of hydrogen adsorbed complexes, metal@C_2_N, and H_2_, respectively. 

For calculation of activation barrier and energy of reaction, Equations (3) and (4) were used, respectively.
E_a_ = E_TS_ − E_Reactant_(3)
ΔE_R_ = E_Product_ − E_Reactant_(4)

In Equation (3), the E_a_ represents activation barrier and E_TS_ represents the energy of transition states. Whereas in Equation (4), the E_R_ shows energy of reaction.

## 3. Results and Discussion

### Geometries and Electronic Properties

The optimized structure of C_2_N consists of hexagonal unit cell (see Figure 1). The C-N bond length is 1.32 Å whereas for C-C bond length is 1.46 Å in the benzene ring and 1.42 Å in the pyrazine ring. The observed C-N-C bond angle is 116.73°. All of these bonding parameters are comparable with already reported in the literature [52].

For each TM being considered, we have studied various spin states in order to obtain thermodynamically the stable spin state. The most stable spin state geometries are reported in the main manuscript (see Figure 2), while least stable M@C_2_N complexes are given in Appendix A. While the interaction energy values for studied TM@C_2_N clusters are reported in Table 1.

Optimized structure of Fe@C_2_N cluster is presented in Figure 2a. Fe atom binds with neighboring nitrogen atoms. The stabilization or adsorption energy in case of Fe@C_2_N cluster is −3.19 eV. The geometry of Co@C_2_N surface is given in Figure 2b, Co shows interaction with neighboring nitrogen atoms and the calculated stabilization energy is −1.42 eV, whereas adsorption of Ni atom over C_2_N surface resulted in the interaction energy of −2.51 eV (Figure 2c). The highest interaction energy value is observed in case of Fe@C_2_N among all studied M@C_2_N clusters, which is attributed to least interaction distance between Fe and N atoms of C_2_N surface. The bonding distance between Fe and nitrogen atoms of C_2_N is 2.53 Å, as compared to 2.64 Å and 2.66 Å for Ni@C_2_N and Co@C_2_N, respectively. Furthermore, it is observed in studied complexes that as the number of unpaired electrons (d-orbital of TMs) decreases, decrease in their interaction energy is observed. Moreover, no distortion is observed in M@C_2_N clusters upon adsorption of TMs (Fe, Ni & Co) due to fused rings of benzene and pyrazine. In addition, no change in bond lengths of C-N and C-C are observed upon adsorption of TMs on C_2_N surface.

## 4. HOMO-LUMO and DOS Analysis

HOMO-LUMO analysis and DOS spectra have been investigated to fully understand the corresponding changes in electronic properties. The charge transfer results and the HOMO-LUMO energy gap of metal doped C_2_N complexes are reported in Table 1. Density of states graphs are presented in Figure 3, which show the formation of new states causes the change in H-L gap. In the DOS graph of Ni doped on C_2_N, the formation of new HOMO states also confirms the change in H-L energy gap.

Upon adsorption of metal atoms, the energy gap (E_H-L_) is significantly reduced. Least reduction in energy gap is observed in case of Co@C_2_N complex which is from 5.61 eV to 5.11 eV. However, a significant decrease in energy gap is observed for Fe@C_2_N and Ni@C_2_N complexes as compared to bare C_2_N surface i.e., E_H-L_ values are 2.56 eV and 1.50 eV, respectively. The change in electronic parameters is confirmed through DOS analysis which clearly shows the formation of new states. Thus, it also explains the significant lowering of HOMO-LUMO energy gap in Fe@C_2_N and Ni@C_2_N complexes. 

In case of Ni@C_2_N complex, potential decrease in H-L gap is observed due to increase in HOMO and decrease in LUMO energies as compared to bare C_2_N surface. Same type of observations is observed from TDOS spectra of Ni@C_2_N due to increase in HOMO energy and decrease in LUMO energy.

## 5. Natural Bond Orbital (NBO) Analysis

NBO analysis was performed on studied TM doped C_2_N clusters to investigate the transfer of charge between C_2_N surface and TM atoms. The values of NBO charges are reported in Table 1. The adsorption of Fe on C_2_N surface resulted in a net charge of 1.43e^−^ on Fe atom. The appearance of positive charge (1.43e^−^) on the Fe atom upon adsorption over C_2_N represents the electron recipient character of C_2_N and electropositive nature of Fe in the most stable geometry of Fe@C_2_N complex. Similarly, in case of Co and Ni dopants, the NBO charges observed are 0.92e^−^ and 0.74e^−^, respectively. In both Co@C_2_N and Ni@C_2_N complexes, the positive sign of charge transfer indicates that charge is shifting towards C_2_N surface from TMs, revealing the electropositive character of studied TMs. Highest charge transfer is observed in case of Fe@C_2_N complex, which reveals the strong interaction among Fe atom and C_2_N support through a charge transfer from Fe atom to C_2_N surface [53]. 

NBO analysis reveals that TMs adsorbed on C_2_N surface showed electropositive character due to their metallic behavior and electron rich C_2_N surface. Highest charge transfer is observed in case of Fe, which verify its high interaction energy with C_2_N surface. However, in case of Co@C_2_N and Ni@C_2_N complexes NBO charges observed are 0.92e^−^ and 0.74e^−^, respectively. 

## 6. Hydrogen Dissociation Reaction on Iron Doped C_2_N Surface

The reaction started with adsorption of H_2_ molecule on C_2_N surface (Figure 4). The hydrogen molecule is adsorbed at iron with adsorption energy of −1.35 eV. The metal atom (Fe1) shows interaction with both hydrogen atoms marked as H2 and H3 with interaction distances of 2.06 Å and 2.07 Å, respectively. Initially H-H bond length of isolated H_2_ is 0.75Å. After adsorption, the hydrogen dissociation proceeds. In the transition state, H-H bond length increases from 0.75Å to 0.93Å and the Metal-Hydrogen bond length decreases to 1.78Å in transition state. Single imaginary frequency confirms that the transition state is located on Fe@C_2_N (see Appendix A for more details). In the product, the Fe-H bond length is 1.67Å and N-H bond length is 1.03Å. The activation barrier for this hydrogen dissociation reaction occurring on Fe@C_2_N is 0.36 eV, while the enthalpy of reaction is −0.05 eV, as mentioned in Table 2.

In case of Fe doped C_2_N catalyst, the activation barrier reduced significantly, which show higher catalytic activity of Fe doped C_2_N catalyst. Iron possesses greater number of unpaired electrons (d-orbital), which are responsible for higher catalytic efficiency of Fe@C_2_N complex.

## 7. Hydrogen Dissociation Reaction on Cobalt Doped C_2_N Surface

In the first step, H_2_ molecule is adsorbed over the C_2_N surface (Figure 5). The stabilization energy observed for the adsorption of hydrogen molecule at cobalt site is −1.93 eV, which is higher than the value observed for adsorption at Fe site (−1.35 eV). Optimized geometry of H_2_ molecules over Co@C_2_N surface reveals that H_2_ is bit tilted. Initially, the interaction distances of Co atom with H2 and H3 atoms of reactant molecule are 1.83 Å and 1.92 Å, respectively (see Figure 5). Then, hydrogen dissociation proceeds through a transition state, where H-H bond length increases from 0.76 Å to 0.85 Å and the Co—H bond length is decreased to 1.76 Å. At final step, the Co—H and N—H bond lengths observed are 1.60 Å and 1.03 Å, respectively. The activation barrier for hydrogen dissociation reaction occurring on Co@C_2_N surface is 0.45 eV, and the enthalpy of reaction is −0.08 eV (see Table 2).

In case of Co@C_2_N complex, the activation barrier of 0.45 eV is observed for hydrogen dissociation, which is comparatively higher as compared to Fe@C_2_N complex (0.36 eV). The higher potential barrier for Co@C_2_N catalyst is due to less unpaired electrons (d-orbital) in TM (Co). 

## 8. Hydrogen Dissociation Reaction on Nickel Doped C_2_N Surface

H_2_ molecule is also adsorbed on Nickel of Ni@C_2_N surface with the adsorption energy of −2.02 eV (see Figure 6). In optimized geometry, reactant hydrogen molecule is oriented almost parallel over the C_2_N surface. The bond distances between nickel atom of Ni@C_2_N and, H2 and H3 atoms of molecule are 1.90 Å and 1.89 Å, respectively, whereas the H—H bond length is 0.76 Å. At transition state, H—H bond length increases from 0.76 Å to 0.84 Å and the Ni—H bond length is decreased from 1.89 Å to 1.75 Å. Finally at product side, the Ni—H bond length gets further reduced to 1.56 Å, whereas N—H bond length is 1.03 Å. In case of Ni@C_2_N cluster, the activation barrier for hydrogen dissociation reaction is 0.40 eV (Table 2), while the enthalpy of reaction is 0.23 eV.

The hydrogen dissociation barrier in Ni@C_2_N complex is 0.40 eV. The observed value of activation barrier in this case is lower than Co@C_2_N catalyst and greater than the Fe@C_2_N catalyst. 

Overall, the order activation barrier observed for studied catalysts is Fe@C_2_N < Ni@C_2_N < Co@C_2_N. The observed trend is quite similar with the trend of TMs doped Al_2_O_3_ reported by Yang et al. [54] for the oxidation of CO by single atom catalysis. 

For comparison, the activation barrier of hydrogen dissociation in our work and some other surfaces are reported in Table 3. Our results show good agreement with already reported values of dissociation barrier using noble TMs. In our case, the lowest hydrogen dissociation barrier is observed for Fe@C_2_N catalyst (0.36 eV), and the value is much better than the reported value of Au/TiO_2_ complex (0.54 eV). Our results are in accordance with the already reported values of dissociation barriers obtained on different surfaces doped with noble TMs. In our case, Fe-incorporated C_2_N surface displays the smallest activated barrier (0.36 eV), which is due to the presence of strong interaction between the metal d orbitals and molecular orbital of H_2_.

## 9. Conclusions

Herein, we have theoretically investigated the hydrogen dissociation reaction on TMs doped C_2_N surface through single atom catalysis. Single atom catalysis provides better efficiency and stability in heterogeneous catalysis. Stable spin states of TMs@C_2_N complexes evaluated for catalytic hydrogen dissociation reaction. Electronic properties (NBO, FMO) of the most stable spin state of TMs@C_2_N complexes are further explored. NBO analysis reveals the electropositive character of TMs, thus, significant charge transfer is observed between TMs and C_2_N surface. Hydrogen molecule, primarily adsorbed on metal doped C_2_N surface during dissociation and then heterolytically dissociated between metal and nitrogen atom of C_2_N surface. The mechanistic pathway of hydrogen dissociation reaction shows that Fe@C_2_N complex is the most suitable catalyst for hydrogen dissociation reaction with activation barrier of 0.36 eV compared to Ni@C_2_N (0.40 eV) and Co@C_2_N (0.45 eV) complexes. Our results indicate that the studied TMs@C_2_N complexes significantly decrease in the activation barrier, which speaks volumes about their success. However, the highest reduction in activation barrier is observed in the case of Fe@C_2_N complex, thus can act as a promising catalyst for hydrogen dissociation reaction in single atom catalysis.

## Figures and Tables

**Figure 1 nanomaterials-13-00029-f001:**
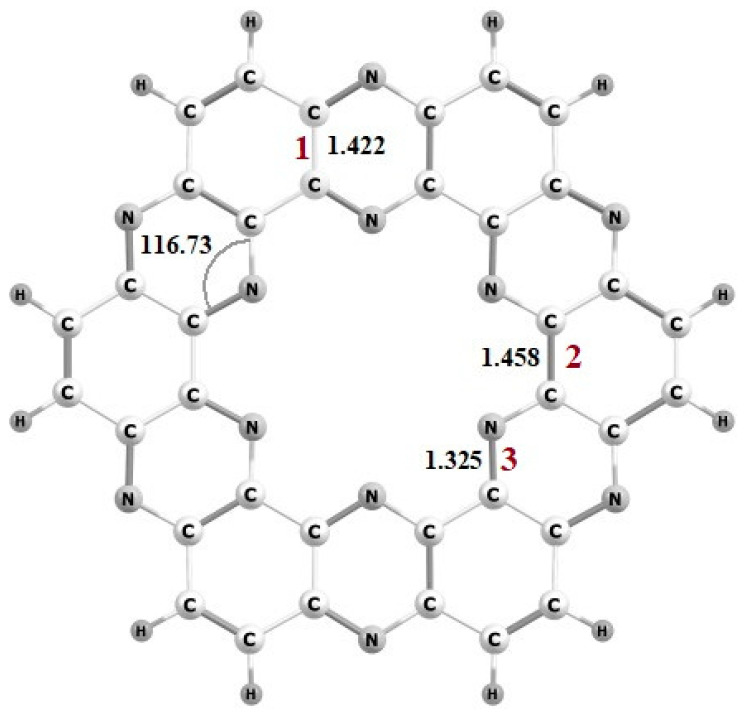
Optimized structure of C_2_N surface.

**Figure 2 nanomaterials-13-00029-f002:**
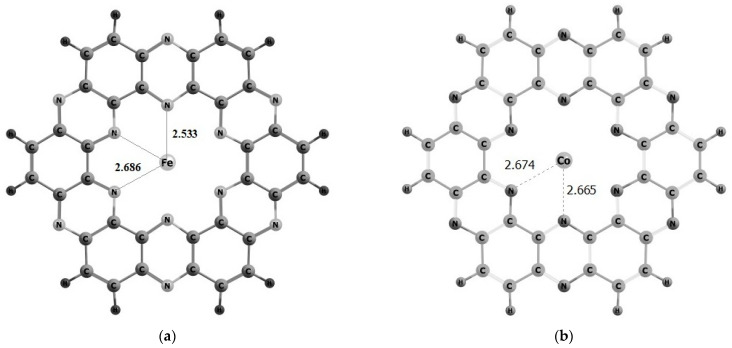
Geometries of (**a**) iron, (**b**) cobalt and (**c**) nickel doped C_2_N complexes.

**Figure 3 nanomaterials-13-00029-f003:**
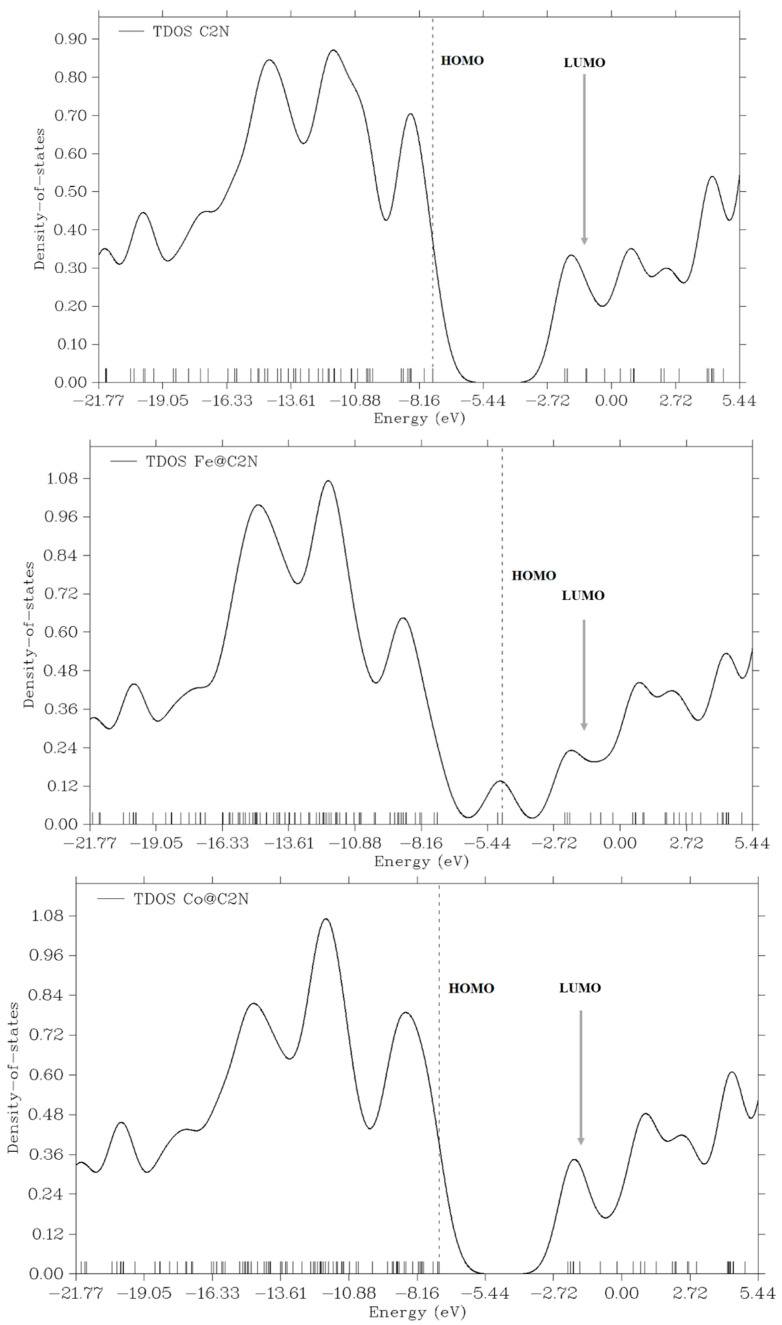
DOS analysis of bare C_2_N surface and metal doped C_2_N surface.

**Figure 4 nanomaterials-13-00029-f004:**
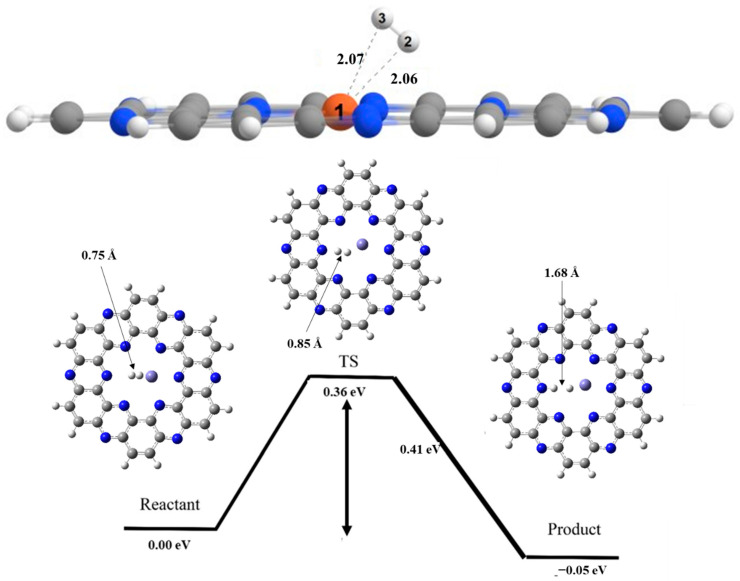
Potential energy surface diagram of H_2_ dissociation on Fe@C_2_N for reactant, transition state and product. Where, grey color is for carbon, white for hydrogen, blue for nitrogen and cobalt blue for iron atom.

**Figure 5 nanomaterials-13-00029-f005:**
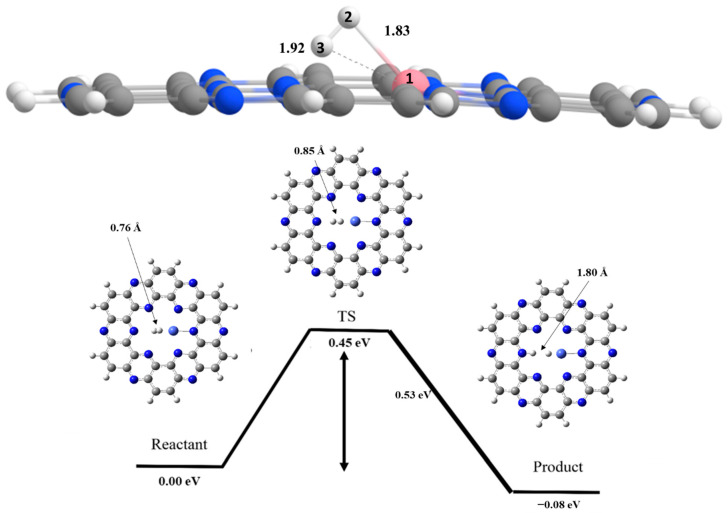
Potential energy surface diagram of H_2_ dissociation on Co@C_2_N for reactant, transition state and product. Where, grey color is for carbon, white for hydrogen, blue for nitrogen and cobalt blue for cobalt atom.

**Figure 6 nanomaterials-13-00029-f006:**
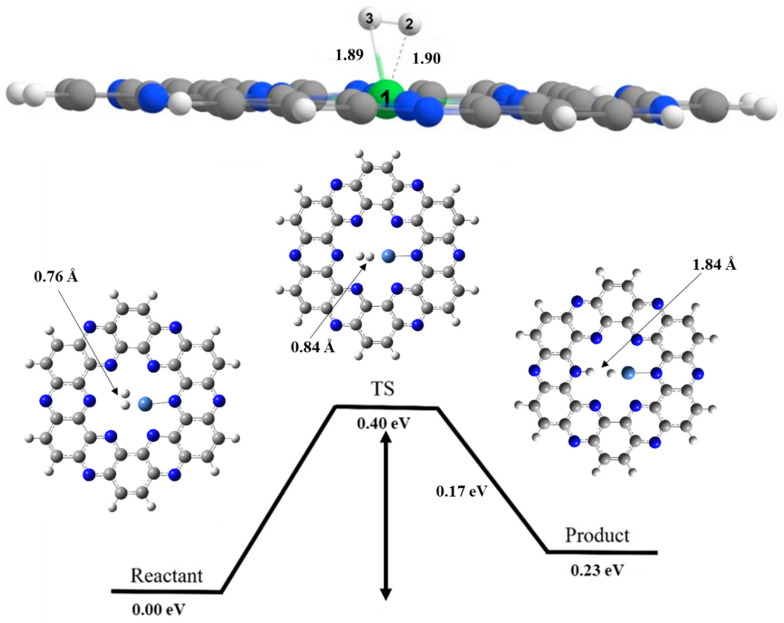
Potential energy surface diagram of H_2_ dissociation on Ni@C_2_N for reactant, transition state and product. Where, grey color is for carbon, white for hydrogen, blue for nitrogen and cobalt blue for nickel atom.

**Table 1 nanomaterials-13-00029-t001:** Interaction energy of metal doped C_2_N surface (eV), M-N distance (Å) between the metal atom and the neighboring nitrogen (N) atoms, charge transfer between the adsorbed metal atom and surface (e^−^) and the HOMO-LUMO energy gap (eV) of M@C_2_N complexes.

M@C_2_N	E_int_	M-N	NBO	HOMO	LUMO	E_H-L_
Fe@C_2_N	−3.19	2.53, 2.69	1.43	−4.84	−2.29	2.56
Co@C_2_N	−1.42	2.66, 2.67	0.92	−7.27	−2.15	5.11
Ni@C_2_N	−2.51	2.64, 2.71	0.74	−4.08	−2.59	1.50
C_2_N	--	--	--	−7.59	−1.99	5.61

**Table 2 nanomaterials-13-00029-t002:** H_2_ dissociation energies at M@C_2_N clusters in eV and calculated bond lengths of H—H bond and M—H bonds, here E_a_ (eV), ΔE (eV) and B.L (Å) represent activation energy barrier, energy of reaction and bond length (Å), respectively.

Reaction Energies		Fe@C_2_N	Co@C_2_N	Ni@C_2_N
E_a_		0.36	0.45	0.40
ΔE		−0.05	−0.08	0.23
H—H Bond length	B.L_R_	0.76	0.75	0.76
	B.L_TS_	0.93	0.85	0.84
	B.L_P_	1.68	1.80	1.84
Ni—H Bond length	B.L_R_	2.06	1.83	1.89
	B.L_TS_	1.78	1.76	1.75
	B.L_P_	1.67	1.60	1.56

**Table 3 nanomaterials-13-00029-t003:** Comparison of current activation barrier of hydrogen dissociation with already reported values over different Surfaces.

Surfaces	Dissociation Barrier	References
Ni adsorbed Mg_17_Al_12_ surface	Mg_16_NiAl_12_ 0.82 eV Mg_15_Ni_2_Al_12_ 0.53 eV	[55]
Ti doped Mg Surface	0.35 eV	[56]
Au/TiO_2_	0.54 eV	[57]
Mg9Rh cluster	0.63 eV	[58]
Fe@C_2_N	0.36 eV	This work
Co@C_2_N	0.45 eV	--
Ni@C_2_N	0.40 eV	--

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
