# Peer review of "Remarkable Single Atom Catalyst of Transition Metal (Fe, Co & Ni) Doped on C2N Surface for Hydrogen Dissociation Reaction"

_nanomaterials, 2022, doi:10.3390/nano13010029_

Round 1

Reviewer 1 Report

In this paper, the authors theoretically investigated the hydrogen dissociation reaction on the cheap TMs-single atom doped C2N catalyst, which shows better efficiency and stability in heterogeneous catalysis. Electronic properties (NBO, FMO) of the most stable spin state of TMs@C2N complexes are further explored, and the significant charge transfer is observed between TMs and C2N surface. The mechanistic pathway of hydrogen dissociation reaction shows that Fe@C2N complex is the most suitable catalyst for hydrogen dissociation reaction with activation barrier compared to Ni@C2N and Co@C2N complexes. The results obtained is interesting, which would give a guidance for the design of high-performance low-cost TMs-single atom doped C2N catalysts. I would like to recommend it for publication in Nanomaterials after minor revision.

The full names of abbreviations should be presented at their first appearance. For instance, the full names of C2N, NBO, and FMO in the abstract part and main text part should be provided.  

Reviewer 2 Report

In this study, Ayub et al. investigated the H2 dissociation reaction theoretically on the transition metal doped C2N surface. The structures of TMs@C2N complexes at different spin states were evaluated for the catalytic H2 dissociation reaction. In addition, the electronic structural properties (NBO, FMO) of the most stable spin state of TMs@C2N complexes were further explored. The results show that Fe@C2N surface is the most suitable catalyst for the H2 dissociation reaction with activation barrier of 0.36 eV compared to those of Ni@C2N (0.40 eV) and Co@C2N (0.45 eV) complexes. This manuscript could be publishable after considering the following issues.

1. In line 188 of page 7, The NBO charges (1.43, 0.92, 0.74 e-) and interaction energies (-3.19, -1.42, -2.51 eV) of Fe@C2N, Co@C2N and Ni@C2N complexes are not strongly related, this is not consistent with the description “The observed charge transfer by TMs (Co & Ni) towards C2N surface are comparatively lower than that of Fe@C2N complex, which is somehow happened due to their less interaction energies with C2N surface”. Please address this problem.

2. Does H2 have more favorable adsorption configurations on Fe@C2N, Co@C2N and Ni@C2N surfaces? Such as the adsorption on top site of metal cluster.

3. Please keep the three structures (reactants, transition states, and products) in the same orientation in Figures 4, 5 and 6.

4. In Figure S1, the optimized geometries of the M@C2N complexes at possible spin states are hardly distinguished, which should be described in more details like adding the M-N bond lengths or the spin densities of the metal atoms.

5. The energy position of the LUMO should be added in Figure 3.

6. There are some small errors, please double check the whole manuscript.

(a) in page 3, the formulas should be aligned with the text.

(b) in line 115 of page 3, the writing form of Emol@C2N-M, EM@C2N and Emol is inconsistent with that of equation (2).

(c) in line 130 of page 3, “Fig. S2” should be changed to “Fig. S1”.

(d) in line 248 of page 10, “Ni-H bonds” should be changed to “M-H bonds”.

(e) please check carefully and unify the format of the references.

Reviewer 3 Report

In this work, Ayub and co-workers report on the theoretical study of hydrogen dissociation supported on metal-coordinated C2N materials. The authors discuss deeply the interaction between the different metals (Co, Fe and Ni) and the lattice showing a higher binding energy for Fe owing to its d orbitals. Subsequently the hydrogen dissociation performance is shown, the activation barriers calculated and compared to the state of the art. The work is interesting, and could be useful for the community of single atoms and electrocatalysis. I can recommend this work to be accepted for publication in Nanomaterials after addressing the next remarks.

The authors mention that there is no change of the N-N bond lengths upon coordination of a transition metal, however there is no direct N-N bonds in C2N, which bond do the authors mean?

C2N possess a wide pore that could host two, or even 3 metal atoms in plane, could the authors elaborate on how the hydrogen dissociation gibs free energy barriers would be affected with a different number of TM atoms in the pore of the C2N lattice?

In the seminal work of Mahmood et al. they show a band gap of 1.7 eV for a C2N covalent organic framework, why do the authors observe values of over 5 eV?. Is it due to the utilization of a molecular building block rather than a periodic C2N framework?

Can the authors elaborate on the oxidation state of the Fe, Co and Ni employed in the calculations?

In the introduction, the authors mentioned that C2N was synthesized experimentally in a “cost effective way”, however the synthesis entails the Schiff base reaction between hexaketocyclohexane and hexaaminobenzene. Hexaaminobenzene is an expensive building block that requires several synthetic steps with explosive reaction intermediates. I would rephrase that by a more descriptive sentence such as providing details of the monomers and the type of reaction, however I wouldn’t say its cost-effective.

While in the literature there are plenty of reports that have shown the diffusion barriers of single atoms in C2N materials, just a few of them have shown the successful synthesis of this materials. These breakthroughs should be mentioned and discussed in the introduction. J. Mater. Chem. A 2022, 10, 6023-6030, Adv. Energy. Mat. 2021, 11, 2003507

In Reference 23, the authors discuss a boron single atom on graphitic carbon nitride for nitrogen reduction reaction. However, the literature on nitrogen reduction is full of false positives in photocatalysis, photo-electrocatalysis and electrocatalysis. I would encourage the authors to remove references towards NRR, or if they discuss about purely theoretical works, to emphasize it

Reviewer 4 Report

Ms. ID.: nanomaterials-2093643

Title: Remarkable single atom catalyst of transition metal (Fe, Co & Ni) doped on C2N surface for hydrogen dissociation reaction

This manuscript presents a theoretical study on the transition metal doped C2N surface through single atom catalysis. The results are relevant. This manuscript is suitable for publishing after minor revisions.

Comments:

1.       Correct some typographical errors:

-          In line 53 change “borsch” by “Bosch”

-          In line 76 change “arenas” by “areas”

2.       The authors should explain in detail the meaning of ETS in equation (3).

3.       The authors should explain the reason why Ni@C2N has lower value of NBO than Co@C2N and the Eint is higher for Ni@C2N than for Co@C2N.

4.       The authors should provide the electronic configuration of Fe, Co and Ni.
